# Diffusional Kurtosis Imaging of White Matter Degeneration in Glaucoma

**DOI:** 10.3390/jcm9103122

**Published:** 2020-09-27

**Authors:** Carlo Nucci, Francesco Garaci, Simone Altobelli, Francesco Di Ciò, Alessio Martucci, Francesco Aiello, Simona Lanzafame, Francesca Di Giuliano, Eliseo Picchi, Silvia Minosse, Massimo Cesareo, Maria Giovanna Guerrisi, Roberto Floris, Luca Passamonti, Nicola Toschi

**Affiliations:** 1Ophthalmology Unit, Department of Experimental Medicine, University of Rome Tor Vergata, 00133 Rome, Italy; alessio.martucci@live.it (A.M.); francesco.aiello@ptvonline.it (F.A.); massimo.cesareo@uniroma2.it (M.C.); 2Neuroradiology Unit, Department of Biomedicine and Prevention, University of Rome Tor Vergata, 00133 Rome, Italy; francescadigiuliano@msn.com; 3San Raffaele Cassino, 03043 Frosinone, Italy; 4Department of Biomedicine and Prevention, University of Rome Tor Vergata, 00133 Rome, Italy; simo85.altobelli@gmail.com (S.A.); francesco2010@hotmail.it (F.D.C.); lanzafame.simona@gmail.com (S.L.); eliseo.picchi@hotmail.it (E.P.); silvia.minosse@libero.it (S.M.); guerrisi@med.uniroma2.it (M.G.G.); toschi@med.uniroma2.it (N.T.); 5Diagnostic Imaging Unit, Department of Biomedicine and Prevention, University of Rome Tor Vergata, 00133 Rome, Italy; roberto.floris@uniroma2.it; 6Institute of Bioimaging and Molecular Physiology, National Research Council, 20090 Milano, Italy; 7Department of Clinical Neurosciences, University of Cambridge, Cambridge CB2 0QQ, UK; 8Athinoula A. Martinos Center for Biomedical Imaging and Harvard Medical School, 149 13th Street, Boston, MA 02129, USA

**Keywords:** primary open angle glaucoma, diffusional kurtosis imaging, diffusion tensor imaging, magnetic resonance imaging

## Abstract

Glaucoma is an optic neuropathy characterized by death of retinal ganglion cells and loss of their axons, progressively leading to blindness. Recently, glaucoma has been conceptualized as a more diffuse neurodegenerative disorder involving the optic nerve and also the entire brain. Consistently, previous studies have used a variety of magnetic resonance imaging (MRI) techniques and described widespread changes in the grey and white matter of patients. Diffusion kurtosis imaging (DKI) provides additional information as compared with diffusion tensor imaging (DTI), and consistently provides higher sensitivity to early microstructural white matter modification. In this study, we employ DKI to evaluate differences among healthy controls and a mixed population of primary open angle glaucoma patients ranging from stage I to V according to Hodapp–Parrish–Anderson visual field impairment classification. To this end, a cohort of patients affected by primary open angle glaucoma (*n* = 23) and a group of healthy volunteers (*n* = 15) were prospectively enrolled and underwent an ophthalmological evaluation followed by magnetic resonance imaging (MRI) using a 3T MR scanner. After estimating both DTI indices, whole-brain, voxel-wise statistical comparisons were performed in white matter using Tract-Based Spatial Statistics (TBSS). We found widespread differences in several white matter tracts in patients with glaucoma relative to controls in several metrics (mean kurtosis, kurtosis anisotropy, radial kurtosis, and fractional anisotropy) which involved localization well beyond the visual pathways, and involved cognitive, motor, face recognition, and orientation functions amongst others. Our findings lend further support to a causal brain involvement in glaucoma and offer alternative explanations for a number of multidomain impairments often observed in glaucoma patients.

## 1. Introduction

Glaucoma, one of the major leading causes of blindness worldwide [1], is an optic neuropathy characterized by death of the retinal ganglion cells and loss of the axons that make up the optic nerve [2]. These alterations are progressive, and if left untreated can produce visual field deficits and optic nerve head atrophy [3]. Visual impairment due to glaucoma affects daily living with wide-ranging implications in functional domains such as driving, walking, and reading [4,5,6]. Moreover, several studies have highlighted a possible connection between reductions of the visual field (VF) and a higher likelihood of developing cognitive impairment, both of which are common phenomena in glaucoma patients [7,8,9,10]. Interestingly, these observations are in keeping with recent findings supporting the hypothesis that glaucoma could be considered to be a neurodegenerative disorder involving the central nervous system (CNS) [4,11,12,13,14,15]. In this context, several studies have highlighted changes in grey matter (GM) [16,17], functional connectivity [14,18,19,20], and white matter (WM) [14,15,20,21] in glaucoma patients as compared with controls. For example, Frezzotti et al. [14] found higher axial diffusivity (AD) in the middle cerebellar peduncle, corticospinal tract (CST), anterior thalamic radiation (ATR), and superior longitudinal fascicle (SLF) in primary open angle glaucoma (POAG) patients relative to controls. These results were confirmed in subsequent studies by the same group [20,21]. Furthermore, Boucard et al. found lower fractional anisotropy (FA) in normal tension glaucoma (NTG) patients as compared with healthy controls, in numerous tracts which were part of the primary visual pathway, such as the optic radiation, as well as in tracts located beyond the visual regions, such as the forceps major (FMa), corpus callosum (CC), and the parietal lobe [15].

Diffusion kurtosis imaging (DKI) is a relatively recent diffusion modeling technique which extends diffusion tensor imaging (DTI). DKI allows a more accurate description of the diffusion signal by adding higher order cumulants to the classical diffusion tensor imaging (DTI) model [22,23,24]. Conceptually, DKI quantifies the deviation from Gaussianity of the self-diffusion probability profile of water molecules in brain tissue. In a complex cytoarchitectonic environment such as white or grey matter, because such deviations are strongly influenced by the underlying microstructure, DKI indices are able to provide complementary information about microstructural alterations as compared with DTI alone [25]. In this context, a small number of previous studies investigated DKI alterations in glaucoma patients using a region-of interest (ROI)-based approach [26,27].

Therefore, the aim of this study was to employ DKI to evaluate normal appearing white matter (NAWM) changes in patients with primary open-angle glaucoma (POAG) as compared with healthy controls in a regionally unbiased, voxel-wise manner. On the basis of prior findings, we hypothesized that group differences would be found both in primary and in secondary tracts. In addition, we posited that differences could also be localized in bundles outside the visual areas, possibly in tracts that interconnect the various lobes, such as the SLF, inferior longitudinal fasciculus (ILF), inferior front-occipital fasciculus (IFOF), and in the uncinate fasciculus (UF).

## 2. Methods

### 2.1. Patient Population

The study’s protocol was approved by our Institutional Ethic Committee and adhered to the tenets of the Declaration of Helsinki. All subjects recruited for this cross-sectional study provided written informed consent. The study was based on the retrospective analysis of prospectively gathered data. A total of, 23 POAG patients and 15 age- and gender-matched healthy subjects were recruited from the glaucoma outpatient clinics at the Department of Ophthalmology of our institution. Patients and controls were both aged between 50 and 80 years (average age 61.3 ± 7.9). All 38 subjects enrolled were right-handed (see Table 1). All subjects underwent a complete ophthalmologic evaluation including administration of a medical history questionnaire, with a focus on local and systemic treatments and family history of glaucoma or neurodegenerative disorders. The evaluation included the following: Best Corrected Visual Acuity (BCVA) examination recorded in logMAR using the Early Treatment Diabetic Retinopathy Study chart (Precision Vision, Woodstock, la Salle, PA, USA); slit lamp examination of the anterior segment; intra ocular pressure measurement using Goldmann applanation tonometry; central corneal thickness evaluation using an ultrasound pachymeter (Pachette DGH500, DGH Technology, Inc., Philadelphia, PA, USA); gonioscopy; 24-2 Swedish Interactive Threshold Algorithms Standard Visual Field (VF) testing; and fundus oculi inspection after pupillary dilation. In order to be included, all participants were required to meet the following inclusion criteria: BCVA > 0.3 logMAR, refractive error < ±5 spherical diopters or < ±3 cylindrical diopters, transparent ocular media, open anterior chamber (Shaffer classification > 20°), and an inter-eye difference not greater than one stage. Exclusion criteria included the following: previous or active optic neuritis, retinal vascular diseases, preproliferative or proliferative diabetic retinopathy, macular degeneration, hereditary retinal dystrophy, use of medication that could affect VF, history of or active neurological, and cerebrovascular or neurodegenerative diseases. The diagnosis of glaucoma was based on the European Glaucoma Society Guidelines.

### 2.2. Visual Field Examination and Patient Groups

VF tests were performed using the Humphrey Visual Field Analyzer with 24-2 Swedish Interactive Threshold Algorithms (Carl Zeiss Meditec Inc., Dublin, CA, USA). Standard Automated Perimetry examinations, as reported in previous studies, were considered to be unreliable and discarded in the presence of fixation losses > 20% and false-positive and -negative errors > 15%. VF estimates were confirmed in at least 3 subsequent VF examinations and were classified according to the Hodapp–Parrish–Anderson criteria [28]. Patients were divided into stages (I–V) according to visual field impairment. Patients within stage 0 according to the Hodapp–Parrish–Anderson (HAP) classification were excluded from the analysis.

### 2.3. Magnetic Resonance Imaging

Magnetic resonance imaging (MRI) was performed using a 3-Tesla scanner (Achieva 3T Intera, Philips Healthcare, Amsterdam, The Netherlands) equipped with gradients of maximum amplitude and rise time of 80 mT/m and 200 mT/m/ms, respectively, and a dedicated 8-channel head coil. The imaging protocol included the following: axial T2-weighted TSE (Turbo Spin Echo) sequence (TR 3000 ms, TE 80 ms, thickness/gap 3 mm/1), axial T2-fluid attenuated inversion recovery (FLAIR) (TR 11000 ms, TE 120 ms, thickness/gap 3 mm/0), sagittal T1-weighted TSE sequence (TR 2000 ms, TE 10 ms, thickness/gap 3 mm/1), and a T1-3D FFE sequence (FOV 224 × 224, TR 25 ms and TE 50 ms), which were employed by an expert neuroradiologist for ruling out visible abnormalities. Patients with recognized abnormalities at morphological imaging, such as evidence of cerebrovascular disease on FLAIR sequences, were excluded from the study. Diffusion-weighed imaging was performed using a spin-echo (SE) echo-planar (EPI) single shot sequence with interleaved slice acquisition and the following parameters: TE 89 ms, TR 7774 ms, slice thickness 2.5 mm, 60 slices, no gap, FOV 240 × 240, matrix 94 × 94 voxel, and SENSE reduction factor R = 2. Diffusion weighting with two distinct b-values (1000 and 2500 s/mm²) was applied in 64 non-coplanar and non-collinear directions. Eight additional non-diffusion-weighted reference images (b0 images) were also acquired.

### 2.4. Image Preprocessing and Model Fitting

Diffusion-weighted images were corrected for subject motion and eddy-current-induced distortions within the ExploreDTI [29] software including geometric image distortion correction and b-matrix reorientation (version 4.8.4 under Matlab 2015, Natick, MA, USA). All DKI metrics were estimated by using the b0 images and complete two-shell data using the DKI model, fitted using constrained nonlinear least squares estimation. From the DKI model, the following parameters were computed: mean kurtosis (MK), kurtosis anisotropy (KA), radial kurtosis (RK), and axial kurtosis (AK). Additionally, we fitted the DTI model to b = 0 and b = 1000 data only and extracted the following metrics from the estimated diffusion tensor: mean diffusivity (MD) and fractional anisotropy (FA).

### 2.5. Statistical Analysis

Voxel-wise statistical analysis of all parameters was carried out using Tract Based Spatial Statistics (TBSS) [30] part of FSL [31]. For each map, we tested the null hypothesis of no differences between healthy controls and all patients, as well as the null hypothesis of no association between all brain metrics and disease stage. All tests were performed through separate general linear models (GLMs). The GLMs included age and gender as nuisance covariates as well as full correction for multiple comparisons over space using permutation-based nonparametric inference within the framework of the general linear model (10,000 permutations). The *p*-values were calculated and corrected for multiple comparisons using threshold-free cluster enhancement TFCE [32] employing the two-dimensional (2D) parameter settings, thereby avoiding the use of an arbitrary threshold for the initial cluster-formation. *p*-values < 0.05 (corrected) was considered to be statistically significant. Anatomical localization of statistically significant clusters according to involved white matter regions and pathways was performed automatically and successively refined manually, in consensus, by two expert neuroradiologists (S.A. and F.G.).

## 3. Results

When comparing all glaucoma patients to healthy controls, we found statistically significant differences in FA (in 1.7% of voxels included in the TBSS skeleton, Figure 1, Appendix A), KA (in 11.4% of voxels included in the TBSS skeleton, Figure 2, Appendix A), MK (in 48.5% of voxels included in the TBSS skeleton, Figure 3, Appendix A) and RK (in 35% of voxels included in the TBSS skeleton, Figure 4, Appendix A). We did not find statistically significant differences in MD and AK metrics. TBSS showed lower FA values in the glaucoma group (GLA) as compared with the controls (CTRL) along and beyond the visual pathway. In particular, we observed lower FA values in several white matter tracts including the genu, the body and the splenium of the CC, in the anterior corona radiata bilateral, in the left superior corona and in the callosal tract (forceps minor, FMi). Statistically significant differences in KA, which again were seen to be lower in GLA as opposed to CTRL, were more diffuse as compared with the differences in FA. They involved the CC (genu, body, and splenium) the posterior thalamic radiation (PTR) bilaterally, including the optic radiation (OR), the left ILF, the left SLF, the left IFOF, in the anterior corona radiata bilaterally, the left superior corona radiata, the left anterior internal capsule, the cingulum and hippocampal part of the cingulate gyrus (CG), the left UF, and the left WM of the temporal and occipital fusiform gyrus. Additionally, in patients with glaucoma, MK values were lower than in the controls in all the same regions where KA differences were observed, with additional and even more widespread differences. In particular, lower values of MK (GLA vs. CTRL) were found in the IFOF bilateral, ILF bilateral, callosal tracts (FMa and FMi), in CST, in the middle cerebellar peduncle, in the anterior and posterior internal capsule and in the WM of the temporal and occipital fusiform gyrus bilateral. Similar results were found for RK. Localizations of white matter clusters in Montreal Neurological Institute (MNI) space are reported in Appendix A
Appendix A. No significant associations were found between DTI/DKI metrics and the disease stage.

## 4. Discussion

In this study, we analyzed NAWM in patients with glaucoma using both DTI and DKI, as well as whole-brain TBSS analysis. Although DKI is not yet routinely used in a clinical setting, it has been shown to be extremely effective in some pathologies including cerebrovascular diseases both in animals and humans [33,34], traumatic brain injury [35,36], brain neoplasms [37,38], aging, and neurodegenerative diseases [39,40], inflammatory brain diseases [41] and in other brain and spine pathologies [42,43], suggesting a growing potential for applications both at the level of experimental animal models as well as in human clinical contexts, whenever it is desirable to evaluate early microstructural alterations (e.g., for diagnosis) or to evaluate the response to pharmacological treatment (clinical routine or clinical trials). A recent DKI study [26] demonstrated mean kurtosis (MK) changes in the visual pathway (optic nerves, lateral geniculate nucleus, optic radiations, and visual cortex) of glaucomatous patients. However, the authors did not investigate white matter regions outside the visual pathway, and their study was based on a priori, manual region of interest (ROI) definition. One other study [27] used a ROI-based approach on Broadman areas 17, 18, and 19 in conjunction with DKI indexes on glaucoma patients, which showed bilateral microstructural changes in the aforementioned areas and highlighted a positive correlation between DKI indexes and abnormal effective connectivity in both primary and secondary visual areas and in the parietal lobe, precuneus and angular gyrus, frontal lobe, as well as the superior and middle frontal gyrus. In our study, both DTI and DKI metrics in NAWM were found to be significantly different in glaucoma patients as compared with the controls in widespread regions which were well beyond the visual areas, and the spatial extent of changes in DKI metrics appeared to be much larger that the spatial extent of changes in DTI metrics. As a matter of fact, we found statistically significant differences between GLA and CTRL in only 1.7% of voxels in FA metrics as compared with the 11.4% of voxels in the KA metrics. Moreover, MD did not show any statistically significant differences between GLA and CTRL, whereas we found them in 48.5% of voxels in the MK metric. Although DKI and DTI share the same lack of specificity in terms of disentangling, for example, demyelination, axonal loss, neuroinflammation or edema, this latter finding could be interpreted as a higher sensitivity of DKI vs. DTI in glaucoma.

The differences observed in our study likely represent a loss of interconnection fibers caused by loss and axonal damage comprising the entire visual pathway and the superior cortices involved in the integration of visual information. The alteration of the interconnection fibers (which support visual information processing) may be at the basis of the mismatch between clinical symptoms and visual field defects that are typical of glaucomatous patients and more evident in early disease stages. In this context, the impairment of peripheral vision only partially explains the difficulty these patients have performing actions that require a visual task (both simple and complex). Interestingly, we found higher KA, MK, and RK in healthy controls as compared with POAG in the ILF, which connects the temporal lobe with the occipital lobe [44,45]. This bundle is the anatomical substrate of the ventral stream (VS) [46,47,48], primarily involved in the recognition of shapes and objects. It originates from the IVα layer of the visual cortex (V1) and reaches the pole of the temporal lobe, receiving inputs mainly from parvicellular neurons. Lesions of this tract are responsible for symptoms such as prosopagnosia. This is compatible with the statistical differences highlighted in the WM of the fusiform gyrus [17,49,50]. The latter is part of the VS and has an important role in object recognition [51], and also in perception and recognition of faces (supported by the so-called fusiform face area) including one’s own face [52,53,54,55]. Interestingly, patients with glaucoma have an impaired face recognition ability [19,56]. Moreover, the left fusiform gyrus also presents an area called visual word form area (VWFA), which is involved in the recognition of visual words and reading [57,58,59]. This could offer an alternative explanation for the reading difficulties experienced by glaucoma patients [5,6,60].

The visual pathway has a second system named the dorsal stream (DS), which plays a key role in guiding movement and in visuo-spatial recognition and coordination [46,48,61,62]. It originates at the level of the IVβ layer visual cortex (V1), ends in the posterior parietal lobe, and uses the SLF as anatomical substrate while maintaining deep connections with the VS along its entire course. Injuries involving the dorsal stream are responsible for symptoms such as simultanagnosia, visual ataxia, spatial hemineglect, acinetopsia, and apraxia. Interestingly, the DKI differences found in our study were distributed along the course of the anatomical substrates of both neural systems mentioned above, with a specific involvement of the SLF, ILF, and IFO.

Furthermore, and perhaps even more interestingly, we found that patients with glaucoma had diffuse involvement of the white matter tracts in the prefrontal cortex [63], as well as in temporo-parietal regions. Although we did not test for cognitive impairment in our population, previous studies [7,8,9,10] have consistently described lower cognitive function in patients with glaucoma which was not merely related to the visual dysfunction per se. The link between glaucoma, cognitive decline, and healthy aging has been recently reported, although a clear pathophysiological framework to explain these associations is still lacking. Our study suggests that the involvement of the non-visual white matter tracts in patients with glaucoma is probably underrecognized and can be mechanistically linked to the increased risk of these patients to express cognitive impairment. Therefore, further research is needed to reveal how the white matter damage may potentially spread from early “bottom-up” visual cortical areas that feed sensory inputs into the brain to more complex, associative, and integrative neural networks that are involved in key cognitive functions such as working memory, attention, and motor control. This has important consequences at the clinical level, for example, to understand a potential risk factor for neurodegenerative disorders such as Alzheimer’s disease. For instance, we highlighted lower KA, MK, and RK value in the FM [15]. This tract is an interhemispheric fiber bundle that links the occipital lobes and also continues to the splenium of the CC. The FM is thought to be involved in higher order visual function such as reading, pattern discrimination, perceptual equivalence, and binocular rivalry [64]. Furthermore, pure topographical disorientation has been observed following a lesion in the FM [65], suggesting that the degeneration of this tract could further support the orientation difficulties experienced by glaucoma patients [5,6,66].

Furthermore, we also found lower MK in the CST [14,20]. This tract is primarily involved in motor function, originates in the primary, the premotor, and supplementary, the cortex connects the latter with the motor neurons of the spinal cord [67]. Furthermore, this could lend support to an additional explanation for the motor difficulties that glaucoma patients experience as highlighted in the work of Trivedi et al. [68]. Moreover, we found a decrease in several metrics (FA, KA, MK, and RK) in several parts of the CC, with exception of the rostrum. The CC is a white matter bundle that connects the two hemispheres and is involved in a vast number of functions [69]. Interestingly, there is evidence of secondary WM degeneration in posterior cortical atrophy (PCA), often called the visual variant of Alzheimer’s disease [70,71,72], and CC degeneration has been hypothesized to constitute as a possible characteristic of the PCA [73].

In this context, while the association between glaucoma and risk to develop Alzheimer’s disease remains a controversial issue, a more focused approach to study the link between cognitive performance over and above visual deficit (e.g., via auditory-based tasks), glaucoma, and changes in the grey and white matter (via different neuroimaging techniques) have the potential to improve our knowledge regarding the true nature of the associations between glaucoma and AD that have been reported in past epidemiological research [4,63,74,75,76].

The Total scan time for the DKI protocol was 9 min and 20 s. However, recent developments in the realm of multiband echo planar imaging are becoming more and more available in clinical scanners. The application of such techniques can reduce this time to below five minutes with a negligible loss of signal-to-noise ratio [77,78]. In this context, the application of DKI in glaucoma, whose social impact is extremely large (more than 60 million patients affected worldwide with a gradual evolution into blindness, i.e., “the silent thief of sight”) is significant because the greatest efforts in the clinical field (diagnosis and monitoring of pharmacological response), as well as research, are concentrated in the very early stages of the disease, while neurodegeneration occurs before visual damage can be objectified by the patient.

Our study is affected by several limitations. We studied a relatively small number of subjects, and therefore our results should be considered to be exploratory. The low sample size is a probable contributor to the null finding when investigating correlation between brain metrics and disease stage directly. To better assess the impact of our findings, a larger sample size and a longitudinal design would be required. Furthermore, the need to employ a larger b value, in the second shell, results in a longer echo time (and subsequently lower signal-to-noise ratio) than what can be achieved in DTI protocols alone. However, it has been shown that even DTI metrics can be estimated with a better accuracy when a DKI term is included [79]. In addition, cognitive testing was not available in our dataset, hence precluding the possibility of formally examining putative associations among glaucoma-related brain changes and cognitive decline.

In conclusion, we have shown widespread normal appearing white matter degeneration in glaucoma which goes well beyond the visual pathways, and whose localization is consistent with a number of multidomain impairments often observed in glaucoma patients. Given that, at present, there are no objective parameters which quantify visual deficit (rather, its estimation relies on a patient’s self-rating while performing a visual test), DKI could be a valuable additional toolfor evaluating the clinical course in a disease which does not commonly involve radiological assessment. Since it has been widely demonstrated that neurodegenerative changes in glaucomatous patients are associated with clinical disease severity, from a purely research perspective, our data also suggest a potential application to standardization and quantification in clinical trials.

## Figures and Tables

**Figure 1 jcm-09-03122-f001:**
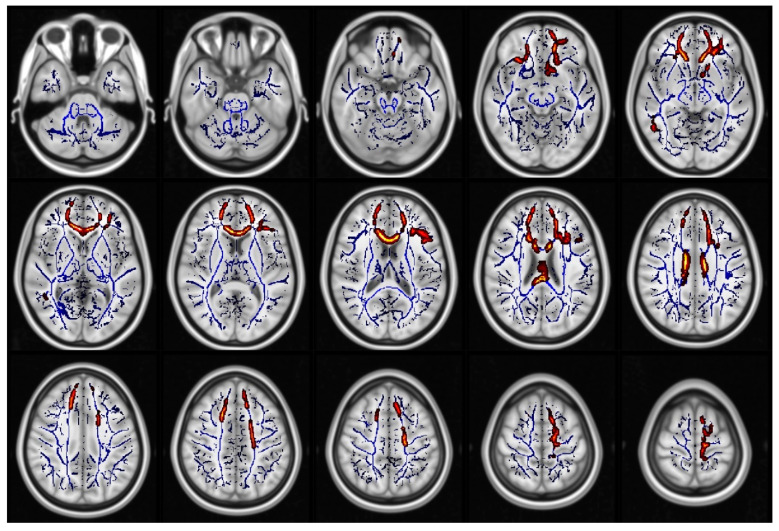
Regions which displayed significant differences in fractional anisotropy (FA) (displayed as red and yellow) (control (CTRL) > glaucoma group (GLA)). The white matter skeleton is shown in blue.

**Figure 2 jcm-09-03122-f002:**
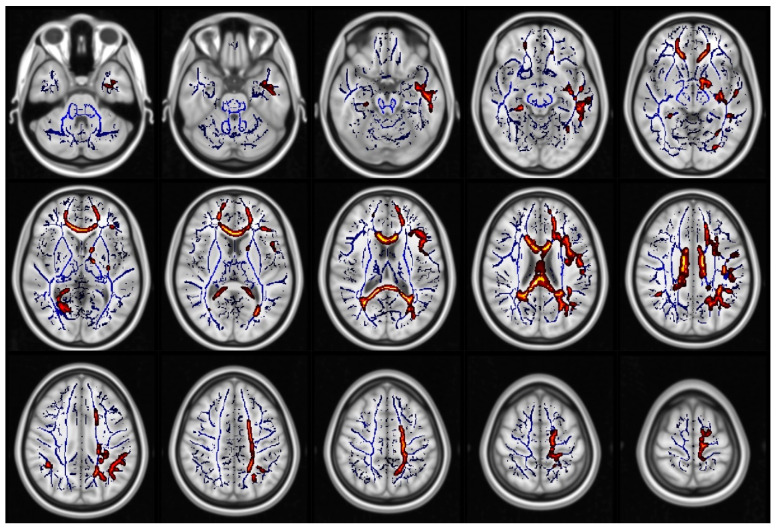
Regions which displayed significant differences in kurtosis anisotropy (KA) (displayed as red and yellow) (CTRL) > glaucoma group (GLA)). The white matter skeleton is shown in blue.

**Figure 3 jcm-09-03122-f003:**
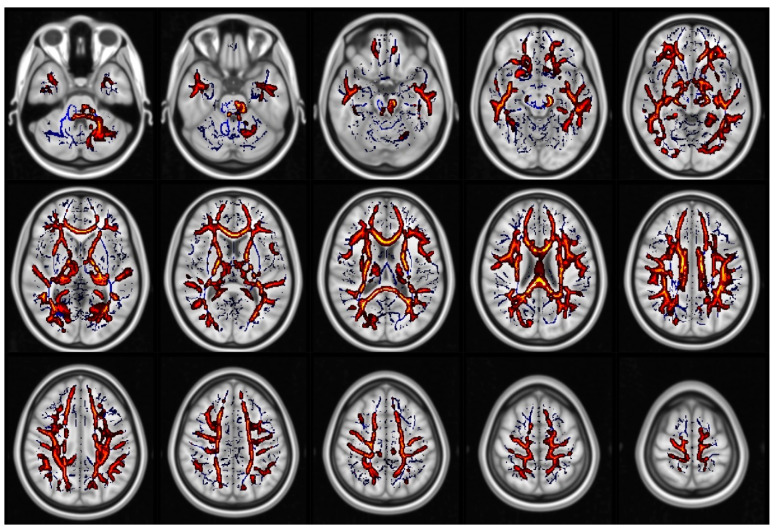
Regions which displayed significant differences in mean kurtosis (MK) (displayed as red and yellow) (CTRL) > glaucoma group (GLA)). The white matter skeleton is shown in blue.

**Figure 4 jcm-09-03122-f004:**
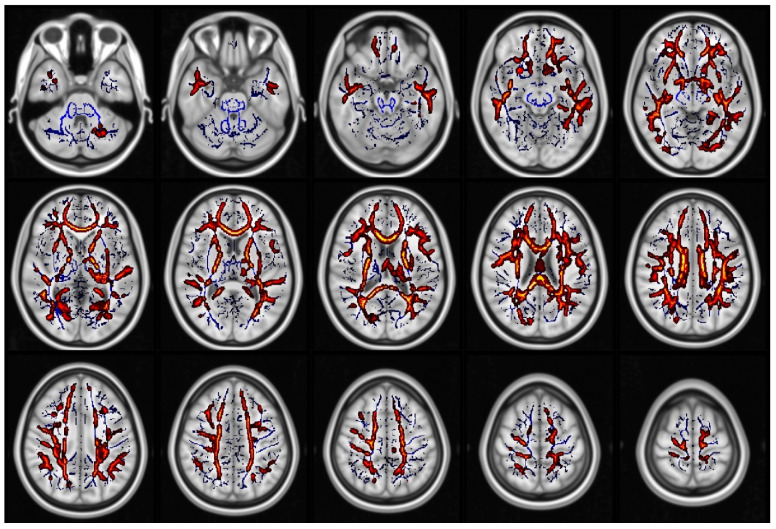
Regions which displayed significant differences in radial kurtosis (RK) (displayed as red and yellow) (CTRL) > glaucoma group (GLA)). The white matter skeleton is shown in blue.

**Table 1 jcm-09-03122-t001:** Study population.

	POAG	Healthy Controls
Population	*n* = 23	*n* = 15
Gender	15 F, 8 M	6 F, 9 M
Average age	61.2 ± 6.9	60.2 ± 9.8
Stages	I: 4, II: 6, III: 6, IV: 5, V: 2	

Primary Open Angle Glaucoma (POAG).

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
