# Peer review of "Diffusional Kurtosis Imaging of White Matter Degeneration in Glaucoma"

_jcm, 2020, doi:10.3390/jcm9103122_

Round 1
Reviewer 1 Report
The study has evaluated the use of Kurtosis imaging (DKI) in patients with glaucoma. The results obtained indicate that this technique, compared to the DTI indices, provides further information about the white matter abnormalities of glaucoma patients.
The study was properly structured. The discussion is relevant and exhaustive. The iconography is of excellent quality and very explanatory. The references are complete.
Reviewer 2 Report
While this paper appears interesting and potentially important, authors are advised to address few issues below before this paper can be considered for publication:
(1) What was the duration of the DKI scans? Can authors discuss in more detail its applicability toward clinical use?
(2) Is there any major advantage of DKI over DTI in detecting glaucoma changes from the current data presented? E.g. Is KA or MK more sensitive or specific than FA? How about AK from DKI vs axial diffusivity from DTI, and RK from DKI vs radial diffusivity from DTI?
(3) Was there any correlation between DKI/DTI metrics and the severity of disease based on the staging classifications or from the clinical ophthalmic assessments performed (e.g. visual field function, corneal thickness etc?)
(4) The authors are advised to discuss how their TBSS results using DTI and DKI are similar to or different from recent TBSS studies in glaucoma with DTI (e.g. PMID 24901535, 27510406, 31578409)
